# Outlining In Vitro and In Silico Cholinesterase Inhibitory Activity of Twenty-Four Natural Products of Various Chemical Classes: Smilagenin, Kokusaginine, and Methyl Rosmarinate as Emboldening Inhibitors

**DOI:** 10.3390/molecules26072024

**Published:** 2021-04-01

**Authors:** F. Sezer Senol Deniz, Gokcen Eren, Ilkay Erdogan Orhan, Bilge Sener, Ufuk Ozgen, Randa Aldaba, Ihsan Calis

**Affiliations:** 1Department of Pharmacognosy, Faculty of Pharmacy, Gazi University, 06330 Ankara, Turkey; fssenol@gazi.edu.tr (F.S.S.D.); bilgesener11@gmail.com (B.S.); 2Department of Pharmaceutical Chemistry, Faculty of Pharmacy, Gazi University, 06330 Ankara, Turkey; gokcene@gazi.edu.tr; 3Department of Pharmacognosy, Faculty of Pharmacy, Karadeniz Technical University, 61080 Trabzon, Turkey; uozgen@ktu.edu.tr; 4Department of Pharmacognosy, Faculty of Pharmacy, Near East University, 99138 Nicosia, Turkey; randaaldaba86@gmail.com (R.A.); ihsan.calis@neu.edu.tr (I.C.)

**Keywords:** natural products, Alzheimer’s disease, cholinesterase inhibition, smilagenin, kokusaginine, methyl rosmarinate, molecular docking

## Abstract

Cholinesterase (ChE) inhibition is an important treatment strategy for Alzheimer’s disease (AD) as acetylcholinesterase (AChE) and butyrylcholinesterase (BChE) are involved in the pathology of AD. In the current work, ChE inhibitory potential of twenty-four natural products from different chemical classes (i.e., diosgenin, hecogenin, rockogenin, smilagenin, tigogenin, astrasieversianins II and X, astragalosides I, IV, and VI, cyclocanthosides E and G, macrophyllosaponins A-D, kokusaginin, lamiide, forsythoside B, verbascoside, alyssonoside, ipolamide, methyl rosmarinate, and luteolin-7-*O*-glucuronide) was examined using ELISA microtiter assay. Among them, only smilagenin and kokusaginine displayed inhibitory action against AChE (IC_50_ = 43.29 ± 1.38 and 70.24 ± 2.87 µg/mL, respectively). BChE was inhibited by only methyl rosmarinate and kokusaginine (IC_50_ = 41.46 ± 2.83 and 61.40 ± 3.67 µg/mL, respectively). IC_50_ values for galantamine as the reference drug were 1.33 ± 0.11 µg/mL for AChE and 52.31 ± 3.04 µg/mL for BChE. Molecular docking experiments showed that the orientation of smilagenin and kokusaginine was mainly driven by the interactions with the peripheral anionic site (PAS) comprising residues of hAChE, while kokusaginine and methyl rosmarinate were able to access deeper into the active gorge in hBChE. Our data indicate that similagenin, kokusaginine, and methyl rosmarinate could be hit compounds for designing novel anti-Alzheimer agents.

## 1. Introduction

Neurodegenerative diseases, most of which are incurable, are characterized by progressive neuronal damage in the brain and peripheral nervous system occurring by complicated heterogenous mechanisms. Among them, Alzheimer’s disease (AD) is a multifactorial neurodegenerative disease defined as the most common type of dementia. In fact, AD has become a major global health concern due to its increasing prevalence particularly in elderly population affecting health care expenses of many countries [1,2]. AD is usually irreversible and abolishes cognitive functions and thinking abilities, while abnormal behavioral changes are also observed in AD patients [3].

Neuropathology of AD is correlated with two main hypotheses generally accepted as “cholinergic hypothesis” and “amyloid hypothesis” [4,5,6]. Currently, the most prescribed drug class for AD treatment is cholinesterase (ChE) inhibitors, e.g., rivastigmine, donepezil, and galanthamine, whereas memantine as another drug option acts through *N*-methyl-D-aspartate (NMDA) receptor antagonism [7,8,9]. ChE comprises two sister enzymes, e.g., acetylcholinesterase (AChE, EC 3.1.1.7) and butyrylcholinesterase (BChE, EC 3.1.1.8), where they are correspondingly responsible for breaking down of acetylcholine (ACh) and butyrylcholine (BCh), the neuromediators transmitting nerve impulses [10]. Since ACh/BCh deficit has been observed in the brains of AD patients, ChE inhibition is an imperative symptomatic treatment strategy towards AD [11]. However, discovery of safer and more effective novel ChE inhibitors are still in need due to side effects of the present ones [12].

Natural products have played a pivotal role in development of new drug candidates since ages. For instance, galanthamine as the latest generation ChE inhibitor is an alkaloid isolated from the bulbs of snowdrop plant (*Galanthus woronowii* Losinsk., Amaryllidaceae) [13,14]. Recently, we also reported *N*-norgalanthamine as a promising dual ChE inhibitor of herbal origin [15]. Similarly, a lot of studies have pointed out to innovation of auspicious natural products towards AD. In the light of those findings, we herein aimed to screen ChE inhibitory capacity of twenty-four natural products from various chemical classes including diosgenin, hecogenin, rockogenin, smilagenin, tigogenin, astrasieversianins II and X, astragalosides I, IV, and VI, cyclocanthosides E and G, macrophyllosaponins A-D, kokusaginin, lamiide, forsythoside B, verbascoside, alyssonoside, ipolamide, methyl rosmarinate, and luteolin-7-*O*-glucuronide using ELISA microtiter assays (Figure 1). The tested natural compounds were randomly selected in general with a special focus on saponosides which have been rarely tested against ChEs. The active inhibitory compounds were further investigated using molecular docking experiments to observe their interactions with the active sites of AChE and BChE.

## 2. Results

### 2.1. Evaluation of ChE Inhibitory Activities of the Tested Compounds

Twenty-four natural products were tested against AChE and BChE at 100 μg/mL. As tabulated in Table 1, smilagenin as steroid derivative and kokusaginine as an alkaloid showed a moderate level of AChE inhibition with IC_50_ values of 43.29 ± 1.38 and 70.24 ± 2.87 µg/mL. Kokusaginine (IC_50_ = 61.40 ± 3.67 µg/mL) and methyl rosmarinate (IC_50_ = 41.46 ± 2.83 µg/mL) were able to inhibit BChE effectively. IC_50_ values galantamine as the reference drug were calculated as 1.33 ± 0.11 µg/mL for AChE and 52.31 ± 3.04 µg/mL for BChE. Rest of the compounds tested had AChE inhibition ranging between none to 25.36 ± 1.11%, while they exerted none to 22.24 ± 2.54% of inhibition against BChE (Table 1).

### 2.2. Molecular Docking Data for the Inhibitory Compounds

In order to analyze the molecular interactions of the compounds, smilagenin, kokusaginine, methyl rosmarinate, hecogenin, tigogenin, and galanthamine were selected to dock into hAChE (PDB: 4EY7) and hBChE (PDB: 4TPK) active site using the Glide module implemented in Schrödinger Small-Molecule Drug Discovery Suite.

According to molecular docking simulations performed with hAChE, the docked compounds were positioned in the bottleneck of the active site gorge interacting with the peripheral anionic site (PAS) comprising residues (Figure 2). Smilagenin exhibiting inhibitory effect against AChE with an IC_50_ value of 43.29 µg/mL, occupied the region between the oxyanion hole and the PAS in AChE active site forming a water-mediated hydrogen bond with TYR124, the PAS residue. Besides, a hydrogen bond stabilizing the inhibitor-protein complex was formed between its 3-hydroxyl group and SER293 (Figure 2A). Kokusaginine having an IC_50_ value of 70.24 µg/mL against AChE was accommodated within the PAS forming water-mediated hydrogen bonds with TYR72, ASP74, TYR124, TRP286, and SER293 as well as π-π stacking contacts with TRP286 and TYR341 (Figure 2B). The best binding pose found for methyl rosmarinate docked into AChE was stabilized by π-π stacking contacts with TRP86 and TRP286 (Figure 2C). Hecogenin and tigogenin showing a weak inhibitory activity to AChE with inhibition% of 25 and 17 at 100 µg/mL bound to the AChE active site by forming water-mediated hydrogen bonds with only the PAS comprising residues, thus leading to occlude the entrance of the active site (Figure 2D,E). The AChE-galanthamine complex was stabilized by several interactions with the active site residues involving ASP74 with a salt bridge, GLU202 with a hydrogen bond, and PHE338 with a π-π stacking contact. Moreover, three π-cation contacts were observed between the nitrogen atom of benzazepine ring and the oxyanion hole residues; TRP86, TYR337, and PHE338 (Figure 2F).

The binding energies of the docked compounds and the residues interacted in hAChE active site were given in Table 2.

In case of the possible binding modes for compounds in hBChE, having a larger catalytic site than AChE allows the compounds access a deeper region in the active gorge as depicted in Figure 3. The proposed binding mode of smilagenine was given in Figure 3A. The compound was located between the acyl binding site and the oxyanion hole of hBChE, close to the catalytic triad via hydrogen bonding between its 3-hydroxyl group and TRP430. Kokusaginine was oriented towards the catalytic triad in hBChE active site interacting HIS438 via water-mediated hydrogen bonding and π-π stacking contacts. Moreover, a water-mediated hydrogen bond with GLU197 and a π-π stacking contact with TRP430 were observed contributing to accommodation within the hBChE active site (Figure 3B). As for methyl rosmarinate displaying inhibitory effect against BChE with an IC_50_ value of 41.46 µg/mL, was positioned toward the region between acyl binding site and oxyanion hole of catalytic site (Figure 3C). The hydrogen bonds formed between hydroxyl groups and the acyl binding site comprising residue SER287 and TYR440 which was located near the choline binding site were found to be main stabilization factors of compound:enzyme complex. Hecogenin and tigogenin accessed the catalytic triad region and interacted with the binding pocket by hydrogen bonding with GLU197 which was located close to catalytic triad members (Figure 3D,E). In case of tigogenin, an additional hydrogen bonding between its 3-hydroxyl group and HIS438 was observed. Galanthamine exhibited a binding mode in BChE active site interacting the PAS residue TYR332 via π-cation contact, while forming a hydrogen bond with HIS438, the catalytic triad member (Figure 3F).

The binding energies of the docked compounds and the residues interacted in hBChE active site were given in Table 3.

## 3. Discussion

Nature has afforded many efficaciously ChE-inhibiting natural products isolated from plants, marine organisms, and microorganisms having different chemical core structures [16,17,18]. In particular, several classes of plant secondary metabolites such as alkaloids, flavonoids, terpenes, coumarins, etc. have been shown to be dual inhibitors of ChEs by our group as well as other researchers [16,19,20,21,22,23,24]. Nevertheless, a very limited number of studies investigated saponin-derivative natural products in terms of ChE inhibition. Thus, several saponin derivatives were included in the current study, only one of which was active inhibitor, e.g., smilagenin, also known as sarsapogenin. Diosgenin, hecogenin, and tigogenin, the common spirostanol derivative steroidal saponins, were found to have a low ChE inhibition in our study. Diosgenin isolated from betel nut was reported to be inhibitory activity against AChE of electric eel origin with have IC_50_ value of 103.60 μg/mL [25]. Consistently with our data on diosgenin, its inhibitory activity could be commented to be quite low, where tannic acid in that study was found to have IC_50_ value of 0.10 μg/mL. On the other hand, only smilagenin closely followed by cannogenin could inhibit AChE. Both of them structurally differ from other steroidal saponins tested herein by carrying hydrogen atom in the alpha position of their 5th carbon. This may have an impact on their ability to inhibit AChE. In another study by Kashyap et al. [26], IC_50_ values of smilagenin (sarsapogenin) isolated from *Asparagus racemosus* was reported 9.9 μM and 5.4 μM for AChE and BChE, respectively, whereas we found only 15.04 ± 3.01% of inhibition by the same compound at 100 μg/mL. Besides, it was revealed to possess neuroprotective effect through anti-amyloidogenic effect against Aβ_42_ and H_2_O_2_-mediated cytotoxicity on PC12 cells. In addition to its ChE inhibitory effect, smilagenin earlier displayed neuroprotection in rat cortical neurons and SH-SY5Y cells [27] and raised stumpy muscarinic ACh receptor density in memory deficit-induced rat brains [28]. Although diosgenin and tigogenin were inactive in our assays, structurally similar solasodine analogues synthesized recently from diosgenin or tigogenin were shown to inhibit AChE [29]. Considering ChE inhibitory activity, structure-activity relationship indicated role of modifications in E and F rings of these compounds and moieties in their A ring. Rockogenin has been tested for the first time against ChEs in the present work.

Astragalosides are the cycloartane type of major saponins in *Astragalus* species, particularly in *A. membranaceus*. During our literature survey, we have not so far come across a study relevant to ChE inhibition by astragalosides. Nevertheless, Santoro et al. lately described low inhibitory activity of three commercially available *A. membranaceus* root extracts along with the extract of the same species they prepared [30]. Their inhibition ranged between none to 27.9 ± 5.0% for AChE and 12.4 ± 7.1% to 27.3 ± 4.0% for BChE, which is in accordance with our data on astragalosides I, IV, and VI. However, it should be also noted that astragalosides were defined to exert neuroprotection by different mechanisms rather than ChE inhibition such as modulation of both phosphoinositide 3-kinase (PI3K)-dependent protein kinase B (PKB, as known as AKT) and extracellular protein kinase (ERK) pathways, inhibition of Aβ_25–35_-induced cytotoxicity and synaptotoxicity as well as blocking mitochondrial dysfunction [31,32]. No neuroprotective effect has been reported for cyclocanthosides and macrophyllosaponins as well as the iridoids tested herein, e.g., lamiide and ipolamide, up to date, where our findings on these compounds constitute the first relevant data.

Verbascoside as a phenylpropanoid derivative isolated from *Olea europea* L. has been demonstrated to show no inhibition against both ChEs [33]. In another study, diminutive inhibitory effect of verbascoside obtained from *Calceolaria talcana* J.Grau & C.Ehr. was reported with IC_50_ values of 189.9 μg/mL for AChE and 105.9 μg/mL for BChE, when compared to galanthamine (reference compound) having consequent IC_50_ values of 13.2 and 7.3 μg/mL against AChE and BChE [34]. In the same study, forsythoside B also possessed trivial BChE inhibition (IC_50_ = 27.6 μg/mL) and no AChE inhibition. Relevantly, verbascoside from *Pseuderanthemum carruthersii* (Seem.) Guill. var. *atropurpureum* (Bull.) Fosb. was identified to have a rather weak AChE inhibition (<50%) at 100 μg/mL [35]. Our previous findings on verbascoside isolated from *Verbascum xanthophoeniceum* Griseb. and *V. mucronatum* Lam. indicated low ChE inhibition below 30% at 100 μg/mL, which comply with the current data on this compound [36,37].

Kokusaginine, a furoquinoline alkaloid tested herein, was identified with moderate AChE and prominent BChE inhibition, closer to that of the reference (Table 1). Previous two reports were consistent with our data on ChE inhibitory ability of kokusaginine isolated from *Esenbeckia leiocarpa* Engl. and *Evodia lepta* (Spreng). Merr. [38,39].

Although rosmarinic acid was earlier reported to be the prominent inhibitor of ChEs by different researchers including us [40,41,42,43], methyl rosmarinate has been assayed for the first time against ChEs in the current work. The molecular docking data was in good agreement with the in vitro results related to BChE inhibition. 

## 4. Materials and Methods

### 4.1. Isolation of the Tested Compounds

Diosgenin (CAS number 512-04-9), hecogenin (CAS number 467-55-0), rockogenin (CAS number 16653-52-4), and smilagenin (CAS number 126-18-1) were obtained from Sigma Aldrich (St. Louis, Missouri, USA). Tigogenin was earlier isolated from the roots of *Digitalis cariensis* Boiss. (Scrophulariaceae). The isolation procedure of tigogenin was described elsewhere [44]. Kokusaginine was previously isolated from the aerial parts of *Haplophyllum myrtifolium* Boiss. (Rutaceae), whose isolation was reported by Sener et al. [45]. Methyl rosmarinate was isolated from *Thymus pseudopulegioides* Klokov & Des. -Shost. (syn. *Thymus nummularius* M. Bieb.) collected from Ormanustu plateau, Macka district, Trabzon province, August, in 2014 at the altitude of 1850 m. The plant was identified by Prof. Dr. Zeki Aytac (Department of Botany, Gazi University, Ankara, Turkey) and preserved at the Herbarium of Pharmacy Faculty of Ankara University (AEF 23176). The air-dried and powdered aerial parts of the plant (300 g) were extracted with methanol (MeOH) (2000 mL × 3) under reflux at 40 °C for 3 h and combined MeOH extracts were concentrated under reduced pressure. MeOH extract (64 g) was suspended with H_2_O:MeOH (9:1, 300 mL). This mixture was partitioned with CHCl_3_ (300 mL × 3) and ethyl acetate (EtOAc) (300 mL × 3), respectively. CHCl_3_, EtOAc, and aqueous phases were evaporated at reduced pressure at 40 °C. EtOAc extract (6.5 g) was subjected to Sephadex LH-20 column by eluting with MeOH. Five fractions (A-E) were obtained. Fraction C (1.3 g) were subjected to silica gel column by eluting with CHCl_3_:MeOH:H_2_O (70:30:3→50:50:5). Fractions 13–45 (160 mg) were combined and then were subjected to VLC. The combined fraction of 96–107 was subjected to Sephadex LH-20 column by eluting with MeOH, which yielded methyl rosmarinate (20 mg). The structure of the compound was elucidated by using by spectroscopic methods using ^1^H-NMR and ^13^C-NMR, which the NMR data of the compound agreed well with the reported data in the literature [46].

Isolation of the tested cycloartane-type triterpene saponins was formerly achieved. Macrophyllosaponins A-D were obtained from the roots of *Astragalus oleifolius* DC. [47], while cyclocanthosides E and G were isolated from the roots of *Astragalus cephalotes* var. *brevicalyx* Eig [48,49,50]. Isolation of astragalosides I, IV, and VI as well as astrasieversianins II and X was described from the roots of *Astragalus melanophrurius* Boiss. [48,50].

### 4.2. Microtiter Assays for Cholinesterase Inhibition

Inhibition of AChE and BChE was determined by Ellman’s method with slight modifications [51]. To the reaction mixture, 140 µL of 0.1 mM sodium phosphate buffer (pH 8), 20 µL of DTNB, 20 µL of enzyme (either AChE or BChE) and 20 µL of the samples to be tested was added later incubated for 15 min at 25 °C. About 10 µL of acetylcholine iodide or butyrylthiocholine chloride which act as substrates to react with DTNB forming 5-thio-2-nitrobenzoate anion, that formed a yellow-colored complex, was added into the incubated sample. The absorbance is measured at 417 nm using a 96-well ELISA microplate reader (VersaMax, Molecular Devices, San Jose, CA, USA) Galanthamine was used the reference drug.

### 4.3. Data Processing for Enzyme Inhibition Assays

The measurements and calculations were evaluated by using Softmax PRO 4.3.2.LS software. Percentage of inhibition of AChE/BChE/TYR was determined by comparison of rates of reaction of test samples relative to blank samples. Extent of the enzymatic reaction was calculated based on the following equation: I% = (C-T)/C × 100, where I% is the activity of the enzyme as percent inhibition. *E* value expresses the effect of the test sample or the positive control on AChE/BChE/TYR enzyme activity articulated as the percentage of the remaining activity in the presence of test sample or positive control. *C* value is the absorbance of the control solvent (blank) in the presence of enzyme, where *T* is the absorbance of the tested sample (plant extract or positive control in the solvent) in the presence of enzyme. Data are expressed as average inhibition ± standard deviation (S.D.) and the results were taken from at least three independent experiments performed in triplicate.

### 4.4. Molecular Docking Experiments

The molecular docking studies were carried out using Glide protocol implemented in the Schrödinger Small-Molecule Drug Discovery Suite (Small-Molecule Drug Discovery Suite 2020-4, Schrödinger, LLC, New York, NY, 2020). The compounds which were built via builder panel in Maestro were subjected to ligand preparation by LigPrep (Schrödinger Release 2020-4: LigPrep, Schrödinger, LLC, New York, NY, 2020) using default conditions. The x-ray crystal structures of the hAChE (PDB: 4EY7) [52] and the hBChE (PDB: 4TPK) [53] were retrieved from the Protein Data Bank. The proteins were prepared using the Protein Preparation Wizard tool. Hydrogen atoms were added followed by assignment of all atom charges and atom types. Finally, energy minimization and refinement of the structures were done up to 0.3 Å RMSD by applying OPLS3e force field. Centroid of the x-ray ligand was defined as the grid box. Van der Waals (vdW) radius scaling factor 1.00, partial charge cutoff 0.25, and OPLS3e force filed were used for receptor grid generation. The compounds prepared by LigPrep were docked into hAChE and hBChE using the extra-precision (XP) docking mode of the Glide without using any constraints and a 0.80 vdW radius scaling factor and 0.15 partial charge cutoff [54]. Best conformation for each compound was chosen based on the lowest XP Glide score.

## 5. Conclusions

According to our results obtained from our screening on twenty-four natural products; smilagenin, kokusaginine, and methyl rosmarinate the compounds came into prominence as the promising ChE inhibitors. The active inhibitors selected for molecular docking studies displayed similarity in interaction with PAS region of hAChE. For BChE, the compounds were located deeply in the catalytic site due to larger active site of hBChE compared to hAChE. The molecular docking simulations performed may suggest that being capable to interact with the key residues highlighted is in full agreement with their inhibitory potency against AChE and BChE. Thus, smilagenin (a saponin derivative), kokusaginine (an alkaloid derivative) and methyl rosmarinate (a phenolic acide ester) out of twenty-four natural products tested herein could be evaluated as the hit molecules for further anti-Alzheimer research. Our research also supports the previous findings that natural compounds having diverse chemical structures still represent a hope for discovering novel candidate ChE inhibitors. 

## Figures and Tables

**Figure 1 molecules-26-02024-f001:**
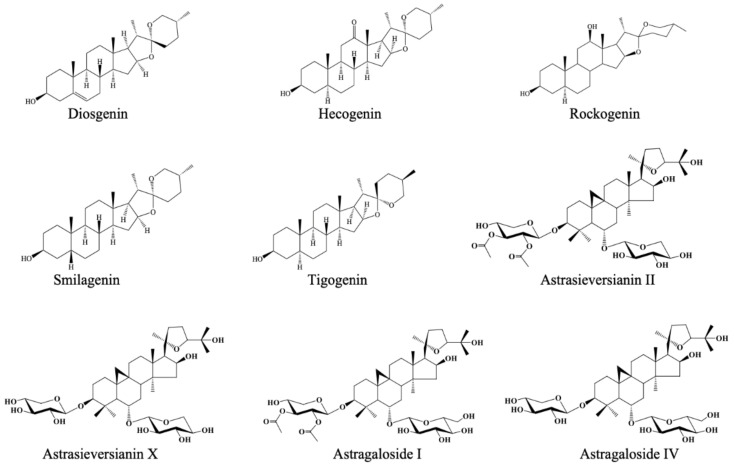
Chemical structures of the tested natural products.

**Figure 2 molecules-26-02024-f002:**
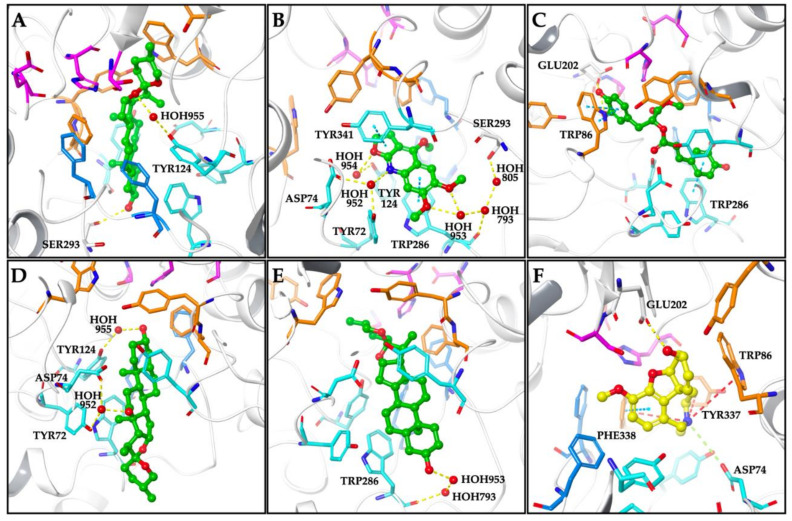
Proposed binding modes for smilagenin (**A**), kokusaginine (**B**), methyl rosmarinate (**C**), hecogenin (**D**), tigogenin (**E**), and galanthamine (**F**) in the hAChE active site (PDB: 4EY7). The compounds are presented as green and yellow ball and stick models. The regions comprising the residues are colored as follows; catalytic triad: magenta, oxyanion hole: orange, acyl binding site: blue, peripheral anionic site: cyan. Hydrogen bonds, π-π stacking contacts, π-cation contacts, and salt bridge are represented by yellow-, cyan-, red-, and green-dashed lines, respectively.

**Figure 3 molecules-26-02024-f003:**
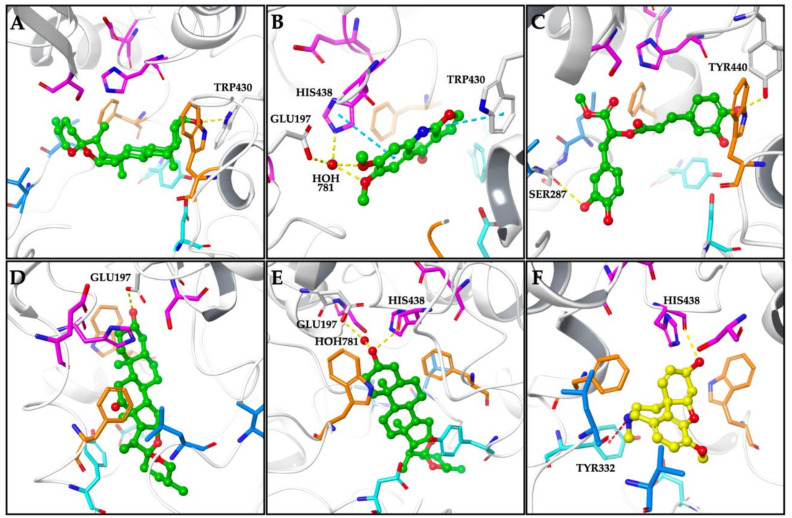
Proposed binding modes for smilagenin (**A**), kokusaginine (**B**), methyl rosmarinate (**C**), hecogenin (**D**), tigogenin (**E**), and galanthamine (**F**) in the hBChE active site (PDB: 4TPK). The compounds are presented as green and yellow ball and stick models. The regions comprising the residues are colored as follows; catalytic triad: magenta, oxyanion hole: orange, acyl binding site: blue, peripheral anionic site: cyan. Hydrogen bonds, π-π stacking contacts, and π-cation contacts are represented by yellow-, cyan-, and red-dashed lines, respectively.

**Table 1 molecules-26-02024-t001:** ChE inhibitory activity (inhibition% ± S.D. and IC_50_ values) of the natural compounds.

Compounds Tested	Chemical Class	ChE Inhibition (Inhibition % ± S.D.^a^ at 100 µg/mL)	
		AChE	BChE
Diosgenin	Steroidal saponins	4.31 ± 0.37	8.12 ± 2.64
Hecogenin		25.36 ± 1.11	17.30 ± 1.84
Rockogenin		5.29 ± 0.52	15.23 ± 2.56
Smilagenin		62.41 ± 2.44 (IC_50_ = 43.29 ± 1.38 µg/mL)	15.04 ± 3.01
Tigogenin		17.09 ± 0.69	22.23 ± 3.09
Astrasieversianin II	Cycloartane triterpenes	8.01 ± 0.48	13.56 ± 1.79
Astrasieversianin X		8.60 ± 0.91	10.06 ± 2.34
Astragaloside I		-^b^	8.38 ± 1.28
Astragaloside IV		4.80 ± 0.05	6.62 ± 2.72
Astragaloside VI		6.23 ± 2.82	7.72 ± 0.53
Cyclocanthoside E		2.18 ± 1.81	5.35 ± 1.82
Cyclocanthoside G		6.85 ± 0.72	8.23 ± 0.24
Macrophyllosaponin A		9.69 ± 0.85	12.17 ± 1.83
Macrophyllosaponin B		13.21 ± 0.92	12.16 ± 1.16
Macrophyllosaponin C		12.42 ± 1.40	16.49 ± 1.57
Macrophyllosaponin D		4.31 ± 1.32	10.10 ± 0.78
Kokusaginine	Alkaloid	62.35 ± 3.20 (IC_50_ = 70.24 ± 2.87 µg/mL)	67.43 ± 3.10 (IC_50_ = 61.40 ± 3.67 µg/mL)
Lamiide	Iridoid	-	8.77 ± 0.92
Ipolamide	Iridoid	-	12.44 ± 2.28
Forsythoside B	Phenylpropanoid	-	14.80 ± 2.31
Verbascoside	Phenylpropanoid	8.12 ± 1.49	22.24 ± 2.54
Alyssonoside	Phenylpropanoid	-	12.99 ± 2.07
Methyl rosmarinate	Phenolic acid ester	16.79 ± 3.84	82.74 ± 0.62 (IC_50_ = 41.46 ± 2.83 μg/mL)
Luteolin-7-*O*-glucuronide	Flavonoid heteroside	-	-
Galanthamine HBr (Reference)		94.19 ± 0.31 (IC_50_ = 1.33 ± 0.11 µg/mL)	60.30 ± 1.36 (IC_50_ = 52.31 ± 3.04 µg/mL)

^a^ Standard deviation (n: 3), ^b^ No inhibition.

**Table 2 molecules-26-02024-t002:** Molecular interactions of the compounds docked with hAChE active site residues.

Compound	Glide Score (kcal/mol)	Interacting Residues and Interaction Types
Smilagenin	−12.10	TYR124 (*HOH955 mediated H-bond*) SER293 (*H-bond*)
Kokusaginine	−10.44	TYR72 (*HOH952 mediated H-bond*) ASP74 (*HOH952 mediated H-bond*) TYR124 (*HOH954 mediated H-bond*) TRP286 (*π-π stack and HOH953, HOH793 mediated H-bond*) SER293 (*HOH953, HOH793, HOH805 mediated H-bond*) TYR341 (*π-π stack*)
Methyl rosmarinate	−10.15	TRP86 (*π-π stack*) TRP286 (*π-π stack*)
Hecogenin	−8.35	TYR72 (*HOH952 mediated H-bond*) ASP74 (*HOH952 mediated H-bond*) TYR124 (*HOH955 mediated H-bond*)
Tigogenin	−7.73	TRP286 (*HOH953, HOH793 mediated H-bond*)
Galanthamine	−11.86	ASP74 (*salt bridge*) TRP86 (*π-cation*) GLU202 (*H-bond*) TYR337 (*π-cation*) PHE338 (*π-cation and π-π stack*)

**Table 3 molecules-26-02024-t003:** Molecular interactions of the compounds docked with hBChE active site residues.

Compound	Glide Score (kcal/mol)	Interacting Residues and Interaction Types
Smilagenin	−7.78	TRP430 (*H-bond*)
Kokusaginine	−7.49	GLU197 (*HOH781 mediated H-bond*) TRP430 (*π-π stack*) HIS438 (*π-π stack and HOH781 mediated H-bond*)
Methyl rosmarinate	−9.39	SER287 (*H-bond*) TYR440 (*H-bond*)
Hecogenin	−8.08	GLU197 (*H-bond*)
Tigogenin	−7.22	GLU197 (*HOH781 mediated H-bond*) HIS438 (*H-bond*)
Galanthamine	−9.51	TYR332 (*π-cation*) HIS438 (*H-bond*)

## Data Availability

Data is contained within the article.

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
