# Peer review of "Outlining In Vitro and In Silico Cholinesterase Inhibitory Activity of Twenty-Four Natural Products of Various Chemical Classes: Smilagenin, Kokusaginine, and Methyl Rosmarinate as Emboldening Inhibitors"

_molecules, 2021, doi:10.3390/molecules26072024_

Round 1

Reviewer 1 Report

In this paper Deniz and coworkers presented the in vitro and in silicon ChE inhibitory activity evaluation of 24 natural products. Overall, the results can supports the conclusions. The quality of the paper can meets the criteria of Molecules. However, the reason why the authors chose these 24 natural products for the ChE activity was not clearly presented in introduction section. The active compounds, similagenin, kokusaginine, and methyl rosmarinate are not lead compounds, but hit compounds.

Author Response

Thank you for your kind comments and evaluation. The selection reason of the tested compounds was expressed as “The tested natural compounds were randomly selected in general with a special focus on saponosides which have been rarely tested against ChEs.” in the introduction.

In similar studies, the active compounds were referred to “lead compounds”. Therefore, we also used this word to describe these compounds. However, taking your suggestion into account, we replaced “lead compounds” by “hit compounds” in the abstract, where it was used only.

Reviewer 2 Report

In this paper, the authors measured the inhibitory power of 24 natural compounds against acetylcholinesterase and butyrylcholinesterase (AChE and BChE, respectively). The results implied that three compounds have promising inhibitory properties; therefore, these three molecules were tested by molecular docking to get insight into the interactions (between the molecule and the active site of the enzyme) leading to the stability of the enzyme-inhibitor complex.

To obtain a more complete picture of the properties of studied compounds, I suggest that authors perform additional molecular docking with some of the other compounds from the set of 24 (not all of them). That would provide the data for making a quantitative correlation between the measured inhibitory activity and the inhibitor-enzyme complex's stability. The reference inhibitor galanthamine should be tested by docking, too.

Regarding reference 25 (Ghayur, M.N.; Kazim, S.F.; Rasheed, H.; Khalid, A.; Jumani, M.I.; Choudhary, M.I.; Gilani, A.H. Identification of antiplatelet 345 and acetylcholinesterase inhibitory constituents in betel nut. Zhong Xi Yi Jie He Xue Bao 2011, 9, 619-625), the full text can be found at https://www.researchgate.net/publication/51215353_Identification_of_antiplatelet_and_acetylcholinesterase_inhibitory_constituents_in_betel_nut

Figure 1 should be redrawn to become more representable and clearer to the audience; molecules are too small and hardly visible. There is an error in the structure of kokusaginine: the nitrogen atom is replaced by carbon.

In Table 1, there are no data given for the compound luteolin-7-O-glucuronide. Please provide enough space between columns.

In Tables 2 and 3, the Glide score values (referring to the ligand-binding free energy) are expressed in kcal/mol, but it is enough to use numbers with one decimal place only.

In row 117 (page 5), please replace TRY72 with TYR72.

Conclusions should provide a more persuasive rationale about the causes of inhibitory activity of examined molecules.

Author Response

In this paper, the authors measured the inhibitory power of 24 natural compounds against acetylcholinesterase and butyrylcholinesterase (AChE and BChE, respectively). The results implied that three compounds have promising inhibitory properties; therefore, these three molecules were tested by molecular docking to get insight into the interactions (between the molecule and the active site of the enzyme) leading to the stability of the enzyme-inhibitor complex.

Thank you for your detailed evaluation.

To obtain a more complete picture of the properties of studied compounds, I suggest that authors perform additional molecular docking with some of the other compounds from the set of 24 (not all of them). That would provide the data for making a quantitative correlation between the measured inhibitory activity and the inhibitor-enzyme complex's stability. The reference inhibitor galanthamine should be tested by docking, too.

Additional molecular docking simulations were performed with hecogenin, tigogenin and galanthamine using both hAChE and hBChE as suggested. Interacting with the key residues comprising the active site was found to be well correlated with inhibitory activities obtained.

Regarding reference 25 (Ghayur, M.N.; Kazim, S.F.; Rasheed, H.; Khalid, A.; Jumani, M.I.; Choudhary, M.I.; Gilani, A.H. Identification of antiplatelet 345 and acetylcholinesterase inhibitory constituents in betel nut. Zhong Xi Yi Jie He Xue Bao 2011, 9, 619-625), the full text can be found at https://www.researchgate.net/publication/51215353_Identification_of_antiplatelet_and_acetylcholinesterase_inhibitory_constituents_in_betel_nut

Thank you for providing us with the reference 25. Going through this reference, data reported for diosgenin was in accordance with our data on the same compound. This was expressed in our paper as follows; “Diosgenin isolated from betel nut was reported to be inhibitory activity against AChE of electric eel origin with have IC50 value of 103.60 mg/mL [25]. Consistently with our data on diosgenin, its inhibitory activity could be commented to be quite low, where tannic acid in that study was found to have IC50 value of 0.10 mg/mL.

Figure 1 should be redrawn to become more representable and clearer to the audience; molecules are too small and hardly visible.

We would like to keep figure 1. We think that it is important for the readers to see all of the tested compounds’ structures in one place cumulatively. It is for easiness for the readers. According to the journal format, we tried to fit them in the space limited. In printing stage, the technical people can adjust their size. It is a technical issue.

There is an error in the structure of kokusaginine: the nitrogen atom is replaced by carbon.

Thanks for the warning. The missing nitrogen was added in the chemical formula and corrected.

In Table 1, there are no data given for the compound luteolin-7-O-glucuronide. Please provide enough space between columns.

Data was already given for luteolin-7-O-glucuronide in Table 1. Since it was not active at all, it was expressed as “-“ in Table 1.

In Tables 2 and 3, the Glide score values (referring to the ligand-binding free energy) are expressed in kcal/mol, but it is enough to use numbers with one decimal place only.

Docking scores given in Table 2 and 3 were simplified as suggested.

In row 117 (page 5), please replace TRY72 with TYR72.

Typo corrected.

Conclusions should provide a more persuasive rationale about the causes of inhibitory activity of examined molecules.

We think that the conclusion should be concise and shortly underlining results of the study. Therefore, we wrote in this style. However, it was extended a bit more.

Round 2

Reviewer 2 Report

The revised manuscript is acceptable for publication in Molecules.